# PointOBB-v2: Towards Simpler, Faster, and Stronger Single Point Supervised Oriented Object Detection

**Botao Ren**[1*], **Xue Yang**[2*], **Yi Yu**[3*], **Junwei Luo**[4], **Zhidong Deng**[1†]

[1]BNRist, THUAI, Department of Computer Science, Tsinghua University
[2]Department of Automation, Shanghai Jiao Tong University
[3]Southeast University    [4]Wuhan University
Code: https://github.com/taugeren/PointOBB-v2

## Abstract

Single point supervised oriented object detection has gained attention and made initial progress within the community. Diverse from those approaches relying on one-shot samples or powerful pretrained models (e.g. SAM), PointOBB has shown promise due to its prior-free feature. In this paper, we propose PointOBB-v2, a simpler, faster, and stronger method to generate pseudo rotated boxes from points without relying on any other prior. Specifically, we first generate a Class Probability Map (CPM) by training the network with non-uniform positive and negative sampling. We show that the CPM is able to learn the approximate object regions and their contours. Then, Principal Component Analysis (PCA) is applied to accurately estimate the orientation and the boundary of objects. By further incorporating a separation mechanism, we resolve the confusion caused by the overlapping on the CPM, enabling its operation in high-density scenarios. Extensive comparisons demonstrate that our method achieves a training speed $15.58\times$ faster and an accuracy improvement of 11.60%/25.15%/21.19% on the DOTA-v1.0/v1.5/v2.0 datasets compared to the previous state-of-the-art, PointOBB. This significantly advances the cutting edge of single point supervised oriented detection in the modular track.

## 1 Introduction

Oriented object detection is essential for accurately labeling small and densely packed objects, especially in scenarios like remote sensing imagery, retail analysis, and scene text detection, where Oriented Bounding Boxes (OBBs) provide precise annotations. However, annotating OBBs is labor-intensive and costly. Therefore, numerous weakly supervised methods have emerged in recent years, including horizontal bounding box supervision and point supervision. Representative methods for horizontal bounding box supervision include H2RBox (Yang et al., 2023a) and H2RBox-v2 (Yu et al., 2023). In addition, point supervision, which only requires labeling the point and category for each object, significantly reduces the annotation cost. Notable point-supervised methods include P2RBox (Cao et al., 2024), Point2RBox (Yu et al., 2024), and PointOBB (Luo et al., 2024).

As illustrated in Fig. 1, existing point-supervised oriented object detection methods can be broadly categorized into three paradigms: (a) SAM-based methods (Cao et al., 2024; Zhang et al., 2024) rely on the powerful SAM (Kirillov et al., 2023) model, which, although effective in natural images, struggles with cross-domain tasks like aerial imagery, particularly in small object and densely packed scenarios. Additionally, SAM-based methods are slow and memory-intensive due to post-processing; (b) Prior-based Weakly-supervised Oriented Object Detection (WOOD) methods, such as Point2RBox (Yu et al., 2024), integrate human priors which reduce generalizability since different datasets require distinct prior knowledge. Further, the end-to-end setup limits flexibility, preventing these methods from leveraging more powerful detectors and benefiting from their performance improvements; (c) Modular WOOD methods (Luo et al., 2024) do not rely on manually designed

---

*Equal contribution. †Corresponding author

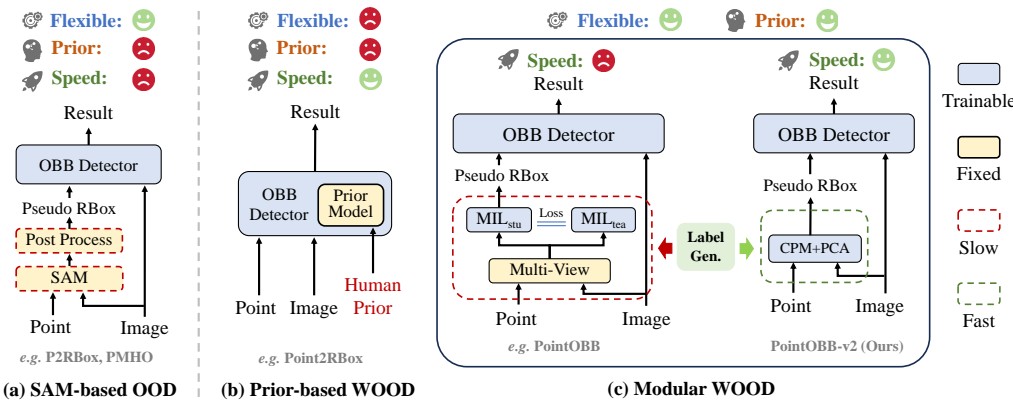

Figure 1: Compare with existing point supervised methods, including (a) Prompt OOD (i.e. SAM based); (b-c) Weakly OOD: Prior-based and Modular. OOD means **O**riented **O**bject **D**etection.

priors and offer greater flexibility by decoupling pseudo-label generation from the detector, making them more suitable for efficient and scalable detection tasks.

As a previous state-of-the-art method, PointOBB falls under the modular WOOD category and offers a feasible solution for point-supervised detection. However, it has several practical limitations: the pseudo-label generation process is very slow, taking approximately 7-8 times longer than the subsequent detector training. Additionally, its training requires significant GPU memory due to multiple view transformations. Moreover, the variability in the number of Region of Interest (RoI) proposals can lead to out-of-memory issues, particularly in dense object scenarios. Although limiting the number of RoI proposals can mitigate this issue, it results in degraded performance.

Considering the aforementioned issues, our motivation is to design a simpler, faster, and stronger method, which leads to the development of PointOBB-v2. Our approach aims to retain the strengths of the modular WOOD paradigm while addressing the inefficiencies of PointOBB, particularly in terms of speed and memory consumption, making it more suitable for real-world applications.

PointOBB-v2 introduces a novel and concise pipeline that discards the teacher-student structure, achieving significant improvements in the accuracy and generation speed of the pseudo-label, and improving memory efficiency, especially in scenarios with small and dense objects. Specifically, we generate Class Probability Maps (CPM) from point annotations and design a novel sample assignment strategy to capture object contours and orientations from the CPM. Next, we apply non-uniform sampling based on the probability distribution and use Principal Component Analysis (PCA) to determine object boundaries and directions. To address dense object distributions, we design a separation mechanism to reduce confusion in pseudo-label generation caused by connected CPMs.

Experimental results demonstrate that our method consistently improves accuracy, speed, and memory efficiency across various datasets compared to PointOBB, achieving several state-of-the-art results. Specifically, on the DOTA-v1.0 dataset, our method, when using pseudo-labels for training with Rotated FCOS, improves the mAP from 30.08% (PointOBB) to 41.68%, a gain of 11.60% mAP. In more challenging datasets like DOTA-v1.5 and DOTA-v2.0, which contain a higher density of small objects, our method achieves mAP of 36.39% and 27.22%, with respective gains of 25.15% and 21.19% over PointOBB, demonstrating its robustness in handling small and densely packed objects. Furthermore, our pseudo-label generation process is 15.58 times faster, reducing the time from 22.28 hours to 1.43 hours. On the DOTA-v1.5 and DOTA-v2.0 datasets, PointOBB requires limiting the number of RoI proposals due to high memory consumption, while our method operates without such restrictions, with a memory usage of approximately 8GB.

Our contributions are summarized as follows:

- We propose a novel and efficient pipeline for point-supervised oriented object detection, which eliminates the time- and memory-consuming teacher-student structure, significantly improving pseudo-label generation speed and reducing memory usage.

- Without any additional deep network design, our method relies solely on class probability maps to generate accurate object contours, using efficient PCA to determine object directions and boundaries. We also design a vector constraint method to distinguish small objects in dense scenarios, improving detection performance.

- Experimental results show that our method consistently outperforms PointOBB across multiple datasets, achieving 11.60%/25.15%/21.19% gain on the DOTA-v1.0/v1.5/v2.0 datasets, with a 15.58× speedup in pseudo-label generation and memory usage reduced to approximately 8GB without limiting RoI proposals.

## 2 RELATED WORK

In addition to horizontal detection (Zhao et al., 2019; Liu et al., 2020), oriented object detection (Yang et al., 2018; Wen et al., 2023) has received extensive attention. In this section, we first introduce oriented detection supervised by rotated boxes. Then, approaches to point-supervised oriented detection and other weakly-supervised settings are discussed.

### 2.1 RBOX-SUPERVISED ORIENTED DETECTION

Representative works include anchor-based detector Rotated RetinaNet (Lin et al., 2020), anchor-free detector Rotated FCOS (Tian et al., 2019), and two-stage solutions, e.g. RoI Transformer (Ding et al., 2019), Oriented R-CNN (Xie et al., 2021), and ReDet (Han et al., 2021). Some research enhances the detector by exploiting alignment features, e.g. $R^3$Det (Yang et al., 2021b) and $S^2$A-Net (Han et al., 2022). The angle regression may face boundary discontinuity and remedies are developed, including modulated losses (Yang et al., 2019a; 2022; Qian et al., 2021) that alleviate loss jumps, angle coders (Yang & Yan, 2020; Yang et al., 2021a; Yang & Yan, 2022; Yu & Da, 2023) that convert the angle into boundary-free coded data, and Gaussian-based losses (Yang et al., 2021c;d; 2023b;c; Murrugarra-Llerena et al., 2024) transforming rotated bounding boxes into Gaussian distributions. RepPoint-based methods (Yang et al., 2019b; Hou et al., 2023; Li et al., 2022) provide alternatives that predict a set of sample points that bounds the spatial extent of an object.

### 2.2 POINT-SUPERVISED ORIENTED DETECTION

Recently, several methods for point-supervised oriented detection have been proposed: **1)** P2RBox (Cao et al., 2024), PMHO (Zhang et al., 2024) and PointSAM (Liu et al., 2024) propose oriented object detection with point prompts by employing the zero-shot Point-to-Mask ability of SAM (Kirillov et al., 2023). **2)** Point2RBox (Yu et al., 2024) has introduced a novel end-to-end approach based on knowledge combination in this domain. **3)** PointOBB (Luo et al., 2024) achieves point annotation based RBox generation method for oriented object detection through scale-sensitive consistency and multiple instance learning.

Among these methods, P2RBox, PMHO and PointSAM require SAM model pre-trained on massive amounts of labeled data, whereas Point2RBox requires one-shot samples (i.e. human priors) for each category. Although achieving better accuracy, they are not as general as PointOBB. Hence, we choose PointOBB as our baseline to develop a simpler, faster, and stronger method, PointOBB-v2.

### 2.3 OTHER WEAKLY-SUPERVISED SETTINGS

Compared to the Point-to-RBox, some other weakly-supervised settings have been better studied. These methods are potentially applicable to our Point-to-RBox task setting by using a cascade pipeline, such as Point-to-HBox-to-RBox and Point-to-Mask-to-RBox. In our experiment, cascade pipelines powered by state-of-the-art weakly-supervised approaches are also adopted for comparison. Here, several representative work are introduced.

**HBox-to-RBox.** The seminal work H2RBox (Yang et al., 2023a) circumvents the segmentation step and achieves RBox detection directly from HBox annotation. With HBox annotations for the same object in various orientations, the geometric constraint limits the object to a few candidate angles. Supplemented with a self-supervised branch eliminating the undesired results, an HBox-to-RBox paradigm is established. An enhanced version H2RBox-v2 (Yu et al., 2023) is proposed to

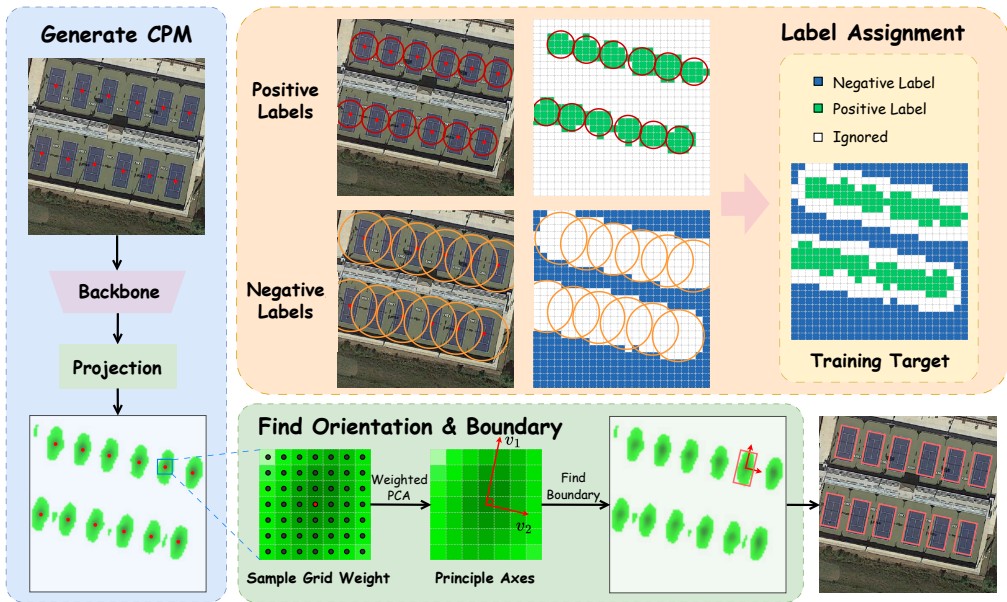

Figure 2: Our PointOBB-v2 first generates a Class Probability Map (CPM) with a positive and negative sample assignment strategy during training. It then applies Principal Component Analysis (PCA) to infer object orientation and boundaries to generate pseudo labels.

leverage the reflection symmetry of objects to estimate their angle, further boosting the HBox-to-RBox performance. EIE-Det (Wang et al., 2024) uses an explicit equivariance branch for learning rotation consistency, and an implicit equivariance branch for learning position, aspect ratio, and scale consistency. Some studies (Iqbal et al., 2021; Sun et al., 2021; Zhu et al., 2023) use additional annotated data for training, which are also attractive but less general.

**Point-to-HBox.** Several related approaches have been developed, including: **1)** P2BNet (Chen et al., 2022) samples box proposals of different sizes around the labeled point and classify them to achieve point-supervised horizontal object detection. **2)** PSOD (Gao et al., 2022) achieves point-supervised salient object detection using an edge detector and adaptive masked flood fill.

**Point-to-Mask.** Point2Mask (Li et al., 2023) is proposed to achieve panoptic segmentation using only a single point annotation per target for training. SAM (Segment Anything Model) (Kirillov et al., 2023) produces object masks from input point/HBox prompts. Though RBoxes can be obtained from the segmentation mask by finding the minimum circumscribed rectangle, such a complex pipeline can be less cost-efficient and perform worse (Yang et al., 2023a; Yu et al., 2023).

## 3 METHOD

Our task focuses on oriented object detection with single point supervision. We first utilize point annotations for each object in the training dataset to generate pseudo labels, which are then used to train an existing detector. As shown in Fig. 2, the model first generates a Class Probability Map (CPM) based on the point annotations. Specifically, during training, we devise a positive and negative sample assignment strategy, where the resulting CPM outlines the rough object contours, with higher probabilities concentrated around the point and along the object axes.

We generate pseudo oriented bounding boxes according to CPM. We perform non-uniform sampling around the point annotation of each object, guided by the probability distribution within the CPM. We convert the sampling process into a weighted probability approach, which maintains the same expected result while eliminating the variance introduced by random sampling. By applying Principal Component Analysis (PCA) to the weighted grid points, we can infer the object's orientation. We then determine the object boundaries by combining the thresholded CPM with the inferred orientation. Furthermore, to address densely populated object scenarios, we introduce a mechanism for differentiating between closely situated objects, ensuring effective separation and accurate detection.

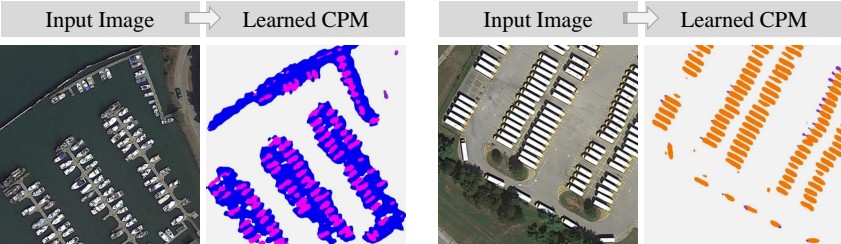

Figure 3: Visualization of class probability map (CPM).

## 3.1 CLASS PROBABILITY MAP GENERATION

The Class Probability Map (CPM) represents the per-class probability for each point on the feature map, with values ranging between [0, 1]. To generate the CPM, our model first takes an image $I$ of dimension $(C, H, W)$ as input and processes it through a ResNet-50 (He et al., 2016) backbone with an FPN (Lin et al., 2017) structure. The final class probability map is derived from the highest-resolution feature map of the FPN, which is then projected through a projection layer. The output is a map of size $(N_{\text{class}}, H_0, W_0)$. Formally, it is defined as:

$$\text{CPM} = \text{Proj}(f(I)_0), \tag{1}$$

where CPM is the class probability map, $\text{Proj}(\cdot)$ represents the projection layer, and $f(I)_0$ is the highest-resolution feature map from the ResNet-50 + FPN.

## 3.2 LABEL ASSIGNMENT

A key component of our approach is the design of a robust sample assignment strategy for both positive and negative samples. This strategy is essential for building an accurate CPM, which outlines rough object contours, concentrating higher probabilities around object centers and along their axes. To ensure reliable separation of objects, especially in densely populated scenarios, our method addresses the challenge of closely situated objects by introducing additional mechanisms for effective differentiation. We illustrate this label assignment in the upper-right part of Fig. 2. The specific details of the sample assignment process are outlined below:

**Positive Label Assignment.** For positive samples, we select all points within a fixed radius $b_1$ (set to 6 in our model) around each point. If a point lies within multiple such radii, it is assigned to the closest center. The condition for positive samples is as follows:

$$\exists \text{GT}_i \in \text{GT}_{1 \sim N}, (d(p, \text{GT}_i) < b_1) \wedge (d(p, \text{GT}_i) = \min(d(p, \text{GT}_{1 \sim N}))) \\ \Rightarrow p \text{ is positive}, \text{cls}(p) = \text{cls}(\text{GT}_i). \tag{2}$$

**Negative Label Assignment.** Given $N$ ground truth objects (GT), for each $\text{GT}_i$, we identify its nearest neighboring object $\text{GT}_j$ based on the Euclidean distance. This gives us a vector $\text{dist}_{min}$ with dimension $[N]$, where each element $\text{dist}_i$ represents the minimum distance between $\text{GT}_i$ and its closest neighbor. We then draw a circle with radius $\alpha \times \text{dist}_i$ around $\text{GT}_i$, where $\alpha$ (set to 1 in our model) is a fixed proportional constant. Points outside all such circles are designated as negative samples. The negative sample condition is formulated as:

$$\forall \text{GT}_i \in \text{GT}_{1 \sim N}, d(p, \text{GT}_i) > \alpha \times \text{dist}_i \Rightarrow p \text{ is negative}. \tag{3}$$

In addition to the above defined negative labels, we also set the middle region between objects as negative to make the boundaries clearer between densely packed objects (denoted as "Neg./M" in ablation Tab. 4). For each $\text{GT}_i$, we identify its nearest neighbor $\text{GT}_j$ that belongs to the same class. A circle is drawn with a radius $b_2$ (set to 4 in our model) and centered at the midpoint of the line connecting $\text{GT}_i$ and $\text{GT}_j$, and points within this circle are assigned as negative samples. The condition is defined as:

$$\forall \text{GT}_i \in \text{GT}_{1 \sim N}, \exists \text{GT}_j \in \text{GT}_{1 \sim N}, d(\text{GT}_i, \text{GT}_j) = \min(d(\text{GT}_i, \text{GT}_{1 \sim N})) \\ \wedge \text{cls}(\text{GT}_i) = \text{cls}(\text{GT}_j) \wedge d(p, (\text{GT}_i + \text{GT}_j)/2) < b_2 \Rightarrow p \text{ is negative}. \tag{4}$$

**Robustness.** While we do not explicitly define positive and negative samples based on precise object contours or oriented bounding boxes, which may result in some inaccuracies during label assignment (i.e. incorrectly assigning a small portion of positive or negative samples during training), this does not significantly hinder our method's ability to learn accurate object contours. These minor label assignment inaccuracies, particularly in densely populated regions or for objects with extreme aspect ratios, do not affect the overall robustness and effectiveness of the approach. As demonstrated in Fig. 3, our strategy is capable of learning the correct contours, even for objects with large aspect ratios and in densely packed scenarios.

## 3.3 Orientation and Boundary estimation via PCA

After obtaining the CPM, we sample points around each ground truth based on the class probabilities, and then apply Principal Component Analysis (PCA) on the sampled points to determine the object's orientation. As shown in bottom part of Fig. 2, we sample points around the GT based on the probabilities in the CPM for the corresponding object class. We choose a $7 \times 7$ grid centered at the GT with the coordinates of the 49 integer points $z_{1 \sim 49}$ as:

$$(x, y), x \in [-3, 3], y \in [-3, 3]. \tag{5}$$

For each grid point $z_i$, we can compute the CPM probability $p_i$ and decide whether to sample the point based on this probability. Once we have the sampled points, we apply PCA to find the primary direction of the point set, which represents the object's orientation. While PCA provides the correct primary direction in expectation, the randomness introduced by sampling can cause variance in the result from a single pass. Although averaging over multiple sampling runs can mitigate this variance, it also increases computational cost.

To address this, we propose an equivalent method that transforms probabilistic sampling into a weighted coordinate transformation. Instead of sampling points probabilistically, we assign a weight of $p_i$ to each point $z_i$, ensuring the same expected outcome while eliminating the variance caused by random sampling. The covariance matrix is then defined as:

$$C_z = \sum_{i=1}^{N} p_i (z_i - \mu_z)^{\mathrm{T}} (z_i - \mu_z). \tag{6}$$

We then perform eigenvalue decomposition on $C_z$:

$$C_z v_i = \lambda_i v_i. \tag{7}$$

The eigenvector $v_1$ corresponding to the largest eigenvalue $\lambda_1$ is chosen as the primary direction. Since $C_z$ is a real symmetric matrix, the secondary direction is guaranteed to be orthogonal to the primary direction. This orthogonality corresponds to the perpendicular relationship between the two adjacent sides of an oriented bounding box.

After identifying the primary and secondary directions, we determine the object boundaries along these directions. Starting from the center, we move along each direction and stop when the value at a position falls below a threshold, indicating the object boundary.

## 3.4 Object Differentiation in Dense Scenarios

In dense scenarios, objects can be difficult to distinguish on the CPM. This can affect the PCA's ability to determine the object orientation and the boundary identification. To address this, we design a "Vector Constraint Suppression" method to resolve boundary ambiguity.

**Vector Constraint Suppression.** Even after determining the correct orientation in dense scenarios, the object boundaries may still be unclear, making it difficult to precisely locate them using the probability threshold described in Sec. 3.3. In most cases, simply distinguishing between two closely positioned objects is sufficient to define the object's boundary.

We propose a simple constraint: For each $\mathrm{GT}_i$, we first find its nearest same-class neighbor $\mathrm{GT}_j$ and compute the vector $u = \langle \mathrm{GT}_i, \mathrm{GT}_j \rangle$ between $\mathrm{GT}_i$ and $\mathrm{GT}_j$. If the angle between this vector and the primary or secondary direction is smaller than a threshold $\alpha$ (set to $\pi/6$ in our model),

Table 1: Results of each category on the DOTA-v1.0 dataset. FCOSR and ORCNN refer to Rotated FCOS and Oriented R-CNN. * indicates using additional **human knowledge** priors.

| Method | PL | BD | BR | GTF | SV | LV | SH | TC | BC | ST | SBF | RA | HA | SP | HC | mAP$_{50}$ |
|---|---|---|---|---|---|---|---|---|---|---|---|---|---|---|---|---|
| Point2Mask-RBox | 4.0 | 23.1 | 3.8 | 1.3 | 15.1 | 1.0 | 3.3 | 19.0 | 1.0 | 29.1 | 0.0 | 9.5 | 7.4 | 21.1 | 7.1 | 9.72 |
| P2BNet+H2RBox | 24.7 | 35.9 | 7.0 | 27.9 | 3.3 | 12.1 | 17.5 | 17.5 | 0.8 | 34.0 | 6.3 | 49.6 | 11.6 | 27.2 | 18.8 | 19.63 |
| P2BNet+H2RBox-v2 | 11.0 | 44.8 | 14.9 | 15.4 | 36.8 | 16.7 | 27.8 | 12.1 | 1.8 | 31.2 | 3.4 | 50.6 | 12.6 | 36.7 | 12.5 | 21.87 |
| Point2RBox-RC | 62.9 | 64.3 | 14.4 | 35.0 | 28.2 | 38.9 | 33.3 | 25.2 | 2.2 | 44.5 | 3.4 | 48.1 | 25.9 | 45.0 | 22.6 | 34.07 |
| Point2RBox-SK* | 53.3 | 63.9 | 3.7 | 50.9 | 40.0 | 39.2 | 45.7 | 76.7 | 10.5 | 56.1 | 5.4 | 49.5 | 24.2 | 51.2 | 33.8 | 40.27 |
| PointOBB (FCOSR) | 26.1 | 65.7 | 9.1 | 59.4 | 65.8 | 34.9 | 29.8 | 0.5 | 2.3 | 16.7 | 0.6 | 49.0 | 21.8 | 41.0 | 36.7 | 30.08 |
| PointOBB-v2 (FCOSR) | 64.5 | 27.8 | 1.9 | 36.2 | 58.8 | 47.2 | 53.4 | 90.5 | 62.2 | 45.3 | 12.1 | 41.7 | 8.1 | 43.7 | 32.0 | 41.68 |
| PointOBB (ORCNN) | 28.3 | 70.7 | 1.5 | 64.9 | 68.8 | 46.8 | 33.9 | 9.1 | 10.0 | 20.1 | 0.2 | 47.0 | 29.7 | 38.2 | 30.6 | 33.31 |
| PointOBB-v2 (ORCNN) | 63.7 | 45.6 | 2.0 | 39.5 | 50.5 | 49.6 | 45.4 | 89.8 | 62.9 | 41.3 | 13.6 | 42.8 | 8.9 | 39.5 | 29.5 | 41.64 |
| PointOBB (ReDet) | 24.2 | 75.0 | 0.5 | 60.6 | 59.3 | 46.1 | 45.7 | 6.1 | 10.1 | 25.0 | 0.2 | 50.4 | 30.0 | 45.0 | 31.1 | 33.95 |
| PointOBB-v2 (ReDet) | 65.4 | 52.1 | 2.2 | 44.4 | 55.0 | 49.3 | 51.8 | 89.0 | 70.2 | 47.0 | 16.2 | 43.9 | 13.0 | 43.8 | 29.4 | **44.85** |

we consider this direction valid for boundary definition. The boundary is then constrained by the following condition:

$$u \cdot v_k < \frac{1}{2} \times d(\text{GT}_i, \text{GT}_j) \quad \text{if angle}(u, v_k) < \alpha \tag{8}$$

where $v_k$ is the primary or secondary direction, $\text{GT}_j$ is the nearest same-class object to $\text{GT}_i$, and $\frac{1}{2}$ means that the boundary should be closer to $\text{GT}_i$ than to $\text{GT}_j$.

# 4 EXPERIMENT

## 4.1 DATASETS

**DOTA** (Xia et al., 2018) is a large-scale dataset designed for object detection in aerial images, covering various object categories and complexities. DOTA has three versions:

**DOTA-v1.0** has 2,806 images with 188,282 instances across 15 categories, annotated as: Plane (PL), Baseball Diamond (BD), Bridge (BR), Ground Track Field (GTF), Small Vehicle (SV), Large Vehicle (LV), Ship (SH), Tennis Court (TC), Basketball Court (BC), Storage Tank (ST), Soccer Ball Field (SBF), Roundabout (RA), Harbor (HA), Swimming Pool (SP), Helicopter (HC). The images range from 800×800 to 4,000×4,000 pixels and exhibit significant variation in scale and orientation.

**DOTA-v1.5** extends DOTA-v1.0 by adding annotations for extremely small objects (less than 10 pixels) and introducing a new category, Container Crane (CC). It includes a total of 403,318 instances while retaining the same image count and dataset split as DOTA-v1.0.

**DOTA-v2.0** further expands the dataset to 11,268 images and 1,793,658 instances, covering 18 categories. Two additional categories, Airport (AP) and Helipad (HP), are introduced, providing a more diverse and challenging set of aerial images.

**RSAR** (Zhang et al., 2025) is a large-scale dataset for rotated object detection in SAR imagery, comprising 95,842 images and 183,534 annotated instances across 6 categories.

**SKU110K-R** (Pan et al., 2020) is a dataset focused on detecting densely packed retail scenes, consisting of 11,762 images and 110,712 annotated objects.

## 4.2 EXPERIMENTAL SETTINGS

Our implementation is based on the MMRotate library (Zhou et al., 2022). In the pseudo-label generation stage, we train the model for 6 epochs using momentum SGD as the optimizer. We set the weight decay to 1e-4, with an initial learning rate of 0.005, which decays by a factor of 10 after the 4[th] epoch. The batch size for training is set to 2. For the detector training phase using pseudo-labels, we use the same detector configurations as the default settings in MMRotate. Throughout the entire training process, random flipping is employed as the only data augmentation technique. Our experiments were accelerated using two GeForce RTX 3090 GPUs.

Table 2: Results on DOTA-v1.0/v1.5/v2.0, RSAR and SKU110K-R, reporting the $\mathbf{mAP_{50}}$ metric. FCOSR and ORCNN refer to Rotated FCOS and Oriented R-CNN. $^*$ indicates using additional **human knowledge** priors.

| Method | DOTA-v1.0 | DOTA-v1.5 | DOTA-v2.0 | RSAR | SKU110K-R |
|---|---|---|---|---|---|
| Point2Mask-RBox | 9.72 | - | - | - | - |
| P2BNet+H2RBox | 19.63 | - | - | - | - |
| P2BNet+H2RBox-v2 | 21.87 | - | - | - | - |
| Point2RBox-RC | 34.07 | 24.31 | 14.69 | - | 3.41 |
| Point2RBox-SK$^*$ | 40.27 | 30.51 | 23.43 | - | - |
| PointOBB (FCOSR) | 30.08 | 10.66 | 5.53 | 13.80 | - |
| PointOBB-v2 (FCOSR) | 41.68 (+11.60) | 30.59 (+19.93) | 20.64 (+15.11) | 18.99 (+5.19) | 56.63 |
| PointOBB (ORCNN) | 33.31 | 10.92 | 6.29 | 12.30 | - |
| PointOBB-v2 (ORCNN) | 41.64 (+8.33) | 32.01 (+21.09) | 23.40 (+17.11) | 22.61 (+10.31) | 62.76 |
| PointOBB (ReDet) | 33.95 | 11.24 | 6.03 | - | - |
| PointOBB-v2 (ReDet) | 44.85 (+10.90) | 36.39 (+25.15) | 27.22 (+21.19) | - | - |

Table 3: Comparison of the training time in pseudo-label generation phase and the accuracy between PointOBB and PointOBB-v2. The reported $\mathbf{mAP_{50}}$ is trained with Rotated FCOS.

| Method | Epochs | Train Hours | $mAP_{50}$ |
|---|---|---|---|
| PointOBB | 24 | 22.28 | 30.08 |
| PointOBB-v2 | 6 | 1.43 | 41.68 |

Table 4: Ablation study with different label assignment strategies.

| Pos. | Neg. | Neg./M | $mAP_{50}$ |
|---|---|---|---|
| ✓ | | | 23.62 |
| ✓ | ✓ | | 44.75 |
| ✓ | | ✓ | 18.21 |
| ✓ | ✓ | ✓ | **44.85** |

## 4.3 Main Results

**Results on DOTA-v1.0**. As shown in Tab. 1, our method achieves state-of-the-art performance compared to the previous leading approaches, i.e. PointOBB and Point2RBox. Specifically, under three different detectors, our method attains $mAP_{50}$ scores of 41.68%, 41.64%, and 44.85%, representing improvements of 11.60%, 8.33%, and 10.90% over PointOBB, respectively. Additionally, when compared to Point2RBox-RC, which does not incorporate human prior knowledge, our approach achieves a substantial gain of 10.78%. Even when compared with Point2RBox-SK, which leverages manual sketches to assist in boundary determination, our method still outperforms it by 4.58%. These results demonstrate the robustness and effectiveness of our approach, even without the need for manual prior knowledge.

**Results on DOTA-v1.5/v2.0.** Both DOTA-v1.5 and DOTA-v2.0 present a higher level of difficulty due to the increased number of densely packed and smaller objects. As Tab. 2 shows, our method demonstrates significant improvements over other approaches on these more challenging datasets, indicating its strength in handling small and densely distributed objects, attributed to the separation mechanism we designed. In comparison to PointOBB, our method achieves substantial gains on both DOTA-v1.5 and DOTA-v2.0, with greater absolute improvements and higher percentages. For instance, when trained with ReDet, our approach improves by 36.39% on DOTA-v1.5 and 27.22% on DOTA-v2.0, corresponding to 25.15% and 21.19% increases, which surpass the 10.90% improvement on DOTA-v1.0. Furthermore, our method consistently outperforms Point2RBox. Even when compared to Point2RBox-SK, which incorporates human prior knowledge, our method achieves improvements of 5.88%/3.79% on DOTA-v1.5/v2.0, respectively.

**Computational Cost.** Our method is highly lightweight, primarily due to its single-branch structure, which eliminates the need for the traditional teacher-student framework. Unlike other methods, we do not require multiple image transformations or consistency constraints within the model. As shown in Tab. 3, the pseudo-label training process for our model takes only 1.43 hours, which is 15.58 times faster than the 22.28 hours required by PointOBB.

In terms of memory consumption, our approach is also more efficient. For dense object scenarios such as DOTA-v2.0, our method uses approximately 8GB of memory, making it suitable for most GPUs. In contrast, PointOBB faces out-of-memory issues when handling such dense scenes, requiring restrictions on the number of RoIs to run properly. However, this limitation severely impacts the detector's performance, resulting in numerous small objects being undetected.

Table 5: Ablation study of sample methods. "Probabilistic" means sample according to the CPM probability.

| Sample | Probabilistic | Weighted |
|---|---|---|
| $mAP_{50}$ | 41.40 | **44.85** |

Table 6: Ablation of vector constraint module to handle dense objects. "w" and "w/o" indicate the "with" and "without" the module.

| Vector Constraint | w/o | w |
|---|---|---|
| $mAP_{50}$ | 27.88 | **44.85** |

Table 7: Ablation study of the sampling size of PCA.

| Size | $mAP_{50}$ |
|---|---|
| 5 | 43.35 |
| 7 | **44.85** |
| 9 | 44.17 |
| 11 | 43.63 |

Table 8: Ablation study of point annotations inaccuracy. We report the pseudo-label generation quality (measured by **mIoU**) and detection performance (measured by **$mAP_{50}$**)

| Point Range | DOTA-v1.0 | | DOTA-v1.5 | | DOTA-v2.0 | |
|---|---|---|---|---|---|---|
| | mIoU | $mAP_{50}$ | mIoU | $mAP_{50}$ | mIoU | $mAP_{50}$ |
| 0% | 45.40 | 44.85 | 43.65 | 36.39 | 42.91 | 27.22 |
| 10% | 42.89 | 42.30 | 41.85 | 34.20 | 41.46 | 25.14 |
| 20% | 40.22 | 38.46 | 39.47 | 30.95 | 39.30 | 23.45 |

Table 9: Comparison of pseudo-label generation quality (measured by **mIoU**) and detection performance (measured by **$mAP_{50}$**) across different datasets. Additionally, we selected three representative categories—Small Vehicle (**SV**), Large Vehicle (**LV**), and Ship (**SH**)—which are abundant and exhibit characteristics of small objects and densely packed distributions. **mean** represents the average value across **all classes** in the dataset.

| Dataset | Method | Memory (GB) | mIoU | | | | $mAP_{50}$ | | | |
|---|---|---|---|---|---|---|---|---|---|---|
| | | | SV | LV | SH | mean | SV | LV | SH | mean |
| DOTA-v1.0 | PointOBB | >24* | 57.06 | 49.78 | 46.59 | 44.88 | 59.32 | 46.06 | 45.71 | 33.95 |
| | PointOBB-v2 | 5.99 | 52.74 | 47.65 | 48.82 | 45.40 | 55.00 | 49.27 | 51.77 | 44.85 |
| DOTA-v1.5 | PointOBB | >24* | 9.69 | 26.58 | 26.49 | 30.65 | 0.01 | 4.15 | 11.40 | 11.24 |
| | PointOBB-v2 | 7.35 | 40.42 | 45.62 | 49.74 | 43.65 | 20.52 | 42.97 | 48.53 | 36.39 |
| DOTA-v2.0 | PointOBB | >24* | 10.01 | 24.45 | 21.33 | 26.63 | 0.49 | 1.22 | 0.36 | 6.03 |
| | PointOBB-v2 | 7.67 | 42.47 | 48.14 | 46.45 | 42.91 | 20.94 | 28.76 | 22.15 | 27.22 |

* The training stage requires constraining to around 70 RoIs; otherwise it results in out-of-memory errors.

## 4.4 ABLATION STUDIES

**Label Assignment.** Tab. 4 demonstrates the impact of our three label assignment strategies on model performance. In these experiments, different label assignment strategies were used to train and generate the CPM. When a specific strategy was applied to define positive and negative samples, the remaining points were ignored during training. We observed that using a simple circular strategy to determine positive samples resulted in only 23.62%. However, by incorporating a more comprehensive strategy to identify negative samples, the performance increased significantly to 44.75%. Furthermore, assigning middle region between objects to negative (denoted as "Neg./M") yielded a slight improvement, raising the mAP to 44.85%. This underscores the crucial role of the negative label assignment strategy, which contributes greatly to the performance gains.

**PCA Sampling Strategy and Size.** We conducted ablation experiments on the PCA sampling strategy and the range of sampling sizes. As shown in Tab. 5, our weighted method for PCA calculation improves accuracy by 3.45% compared to the probabilistic method. We also found that this improvement primarily benefits classes with larger aspect ratios, such as large vehicles and harbors. This is because CPM in elongated objects exhibit significant probability variation along the short axis of their oriented bounding boxes, and probabilistic sample method introduces considerable instability. Additionally, we evaluated the impact of the PCA sampling size. As shown in Tab. 7, our method achieves the best performance when the sampling size is set to 7.

**Vector Constraint.** As shown in Tab. 6, applying the vector constraint significantly improves detection performance. Through further analysis, we found that the improvement is primarily concentrated in dense object categories, such as small vehicles, large vehicles, and ships. In contrast, sparse categories like harbors and swimming pools are almost unaffected. This observation aligns with the motivation behind the design of this module, which especially addresses densely packed object scenarios.

### 4.5 ANALYSIS

**Label Accuracy.** Recognizing the potential inaccuracies in human annotations, where the center point might not be perfectly labeled, we conducted experiments by adding noise to the center points to evaluate the robustness of our model. We selected different thresholds $\sigma$ and calculated the object's scale as $S = \sqrt{wh}$. The center points were randomly shifted along a uniformly sampled direction, with the offset distance drawn from a uniform distribution over the range $[-\sigma S, \sigma S]$. We observed a slight performance decrease as the center points were perturbed. As Tab. 8 shows, the average mAP dropped by only 2.27% with a 10% shift. Despite this decline, our method still significantly outperforms PointOBB, demonstrating the strong robustness of our model.

**Quality of Pseudo Labels.** As shown in Tab. 9, our method consistently outperforms PointOBB in generating pseudo labels, with performance gains increasing in more challenging datasets like DOTA-v1.5 and DOTA-v2.0. Specifically, we observe mIoU improvements of 0.52%, 13.00%, and 16.28% across DOTA-v1.0, DOTA-v1.5, and DOTA-v2.0, respectively. Notably, although the improvement in DOTA-v1.0 is only 0.52%, training the same detector with our pseudo labels yields a nearly 10% increase in mAP compared to PointOBB. As shown in Fig. 5, the first and second columns illustrate that our model learns more accurate object scales, while the third column demonstrates that, unlike PointOBB—which produces overlapping pseudo labels for small vehicles—our method effectively distinguishes between these densely packed objects.

**Dense Object Scenarios.** As shown in Tab. 9, we selected three representative categories—`Small Vehicle` (SV), `Large Vehicle` (LV), and `Ship` (SH)—which are characterized by small and densely packed objects. Datasets like DOTA-v1.5 and DOTA-v2.0 introduce a much larger number of these densely packed objects compared to DOTA-v1.0. In these challenging scenarios, our method significantly outperforms PointOBB. For example, in DOTA-v2.0, our method achieves a mean mIoU of 42.91% and mAP of 27.22%, whereas PointOBB drops to 26.63% and 6.03%, respectively. Visualizations further confirm that our model generates better pseudo labels in dense scenes. In terms of detection results, in the last column of Fig. 5, we show a dense scene with 25 large vehicles, where our method detects all of them, while PointOBB identifies only 15.

**Limitations.** (a) Our method assigns negative samples based on the minimum distance between objects, requiring at least two point annotations per image. In scenarios with extremely sparse objects, it may degrade the performance. (b) Some hyperparameters (e.g. the radius in label assignment) are set based on the dataset. They may require adjustments when facing other scenarios.

## 5 CONCLUSION

In this paper, we introduced PointOBB-v2, a simpler, faster, and stronger approach for single point-supervised oriented object detection. By employing class probability maps and Principal Component Analysis (PCA) for object orientation and boundary estimation, our method improves detection accuracy while discarding the traditional time- and memory-heavy teacher-student structure. Experimental results demonstrate that PointOBB-v2 consistently outperforms the previous state-of-the-art across multiple datasets, achieving a training speed 15.58× faster and accuracy improvements of 11.60%/25.15%/21.19% on the DOTA-v1.0/v1.5/v2.0 datasets, with notable gains in small and densely packed object scenarios. Our method achieves a substantial speedup and accuracy boost while using less memory, showcasing its effectiveness for real-world applications.

## 6 ACKNOWLEDGMENTS

This work was supported in part by the National Natural Science Foundation of China (NSFC) under Grant No. 62176134, by research and application on AI technologies for smart mobility funded by SAIC Motor, and by a grant from the Institute Guo Qiang (2019GQG0002), Tsinghua University.

This work was also supported in part by the National Natural Science Foundation of China (NSFC) under Grant No. 62306069.

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

## 7 APPENDIX

### 7.1 NEGATIVE LABEL ASSIGNMENT

To further compare our dynamic minimum radius circle-based strategy for defining negative samples with the fixed radius circle-based strategy, we trained the CPM using both strategies and visualized the results. As shown in Fig. 4, our method better captures the contours of an object, whereas the fixed radius approach produces a CPM that resembles a circle centered on the object, with less distinct contours. This lack of clarity in the contours makes it difficult for PCA to accurately determine the orientation and boundaries based on the CPM.

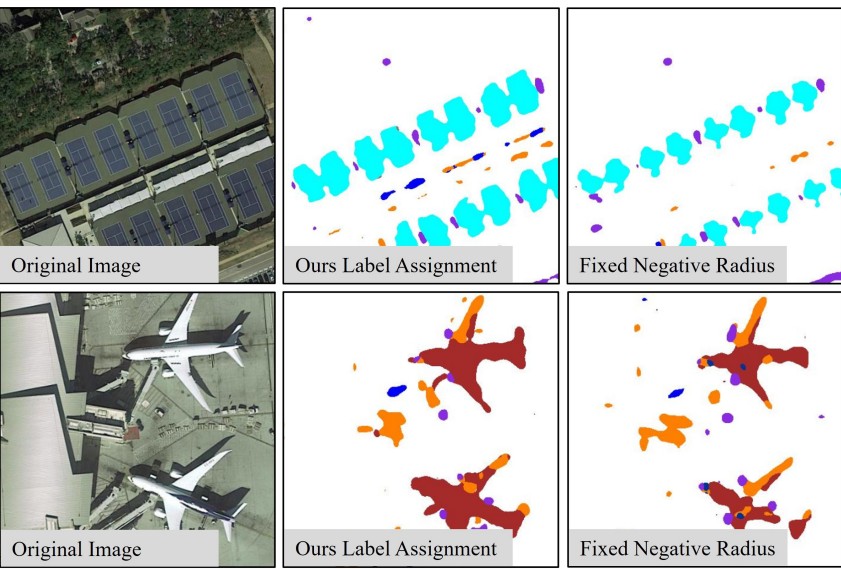

Figure 4: Visualization of the class probability map (CPM). Our dynamic radius negative label assignment vs. fixed radius negative label assignment. Our method produces clearer object contours, whereas the fixed radius approach generates contours that are approximately circular, with the resulting CPM primarily concentrated around the object's center.

### 7.2 ROBUSTNESS OF CPM'S PARAMETER SELECTION

We conducted experiments on the DOTA dataset, focusing on parameters that could potentially impact the results ($\alpha$, b1, b2), and observed minimal fluctuations in the final results. Additionally, the performance consistently exceeded that of PointOBB (33.31%) by a significant margin (minimum 7% gain on Oriented RCNN). Therefore, we conclude that the introduced thresholds do not require precise tuning to achieve strong performance, indicating that the model is not highly sensitive to threshold settings.

Table 10: The performance of our method under different parameter choices.

| Parameters Choice | mAP$_{50}$ (Oriented RCNN) |
|---|---|
| $\alpha, b_1, b_2 = 1, 6, 4$ (Result in Tab. 1) | 41.64 |
| $\alpha, b_1, b_2 = 0.67, 8, 4$ | 41.33 |
| $\alpha, b_1, b_2 = 1, 8, 4$ | 40.68 |
| $\alpha, b_1, b_2 = 1.5, 8, 4$ | 40.22 |
| $\alpha, b_1, b_2 = 1.5, 6, 4$ | **41.95** |
| $\alpha, b_1, b_2 = 0.67, 12, 6$ | 41.74 |

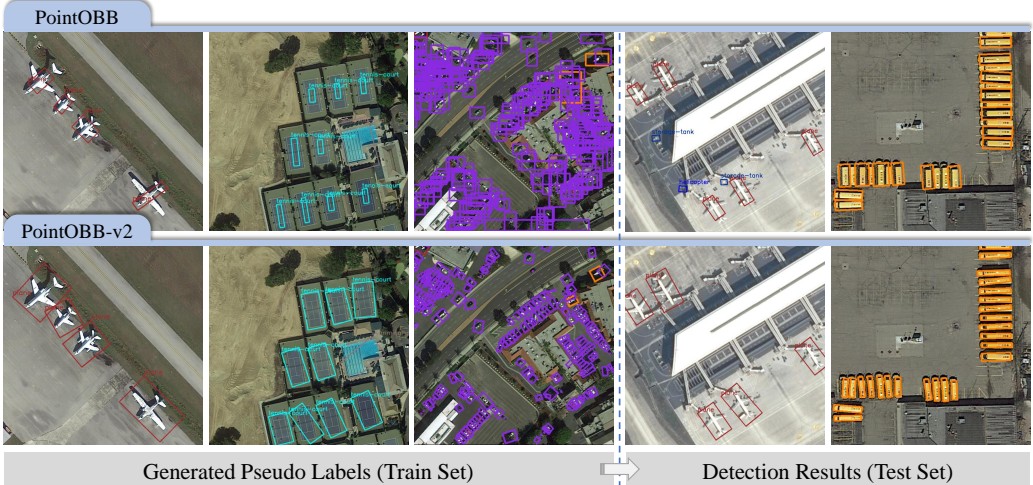

Figure 5: Visualization of pseudo-labels and detection results from our model compared to PointOBB. For clarity, label texts for dense objects are hidden.

## 7.3 VISUALIZATION

We visualize the quality of pseudo labels and detection results generated by our method compared to PointOBB. As shown in Figure 5, our method generates more precise object scales for pseudo labels and performs better in dense object scenarios, where PointOBB often produces overlapping pseudo labels. In terms of detection results, as seen in the fourth column, our approach produces bounding boxes with more accurate scales and fewer false detections. In the last column, we demonstrate a dense scenario with 25 large vehicles, where our method detects all of them, while PointOBB identifies only 15.

## 7.4 DISCUSSION ON ANNOTATION COST VS PERFORMANCE GAP TO FULL-SUPERVISION ACCURACY

In the field of object detection, the cost of point annotations is about 36.5% lower than HBB and 104.8% lower than OBB, and the efficiency of labeling point is significantly higher than both HBB and OBB. This cost analysis has also been adopted and validated in prior works, such as H2RBox (Yang et al., 2023a), PointOBB (Luo et al., 2024), and Point2RBox (Yu et al., 2024), further demonstrating its reliability and relevance.

## 7.5 SOME IMPLEMENTATION DETAILS IN CPM GENERATION MODULE

The activation function used in the middle layers is LeakyReLU, while the last layer employs a sigmoid function. For each point, we first generate its target using a label assignment strategy. Subsequently, we train the CPM using focal loss (Lin et al., 2020).

