# OpenReview forum: "PointOBB-v2: Towards Simpler, Faster, and Stronger Single Point Supervised Oriented Object Detection"
_ICLR.cc/2025/Conference — ICLR 2025 Poster_

### Official Review · Reviewer_oGEA · 2024-11-03

**Soundness:** 3
**Presentation:** 3
**Contribution:** 3
**Rating:** 6
**Confidence:** 4

**Summary:**

This paper begins with a comprehensive review of existing point-supervised methods, examining their relative merits with particular emphasis on the limitations of the state-of-the-art approach, PointOBB in Modular WOOD. The key limitations identified include: 1) inefficient pseudo-label generation, 2) excessive GPU memory consumption, and 3) frequent memory overflow issues during RoI proposal generation.
To overcome these limitations, the authors introduce PointOBB-v2, which presents a streamlined pipeline that eliminates the traditional teacher-student architecture. Instead, it employs class probability maps exclusively for precise object contour generation, incorporating efficient Principal Component Analysis (PCA) for object orientation and boundary determination. Additionally, the study proposes a novel vector constraint mechanism for enhanced detection of small objects in high-density scenarios.
Experimental results reveal substantial improvements in pseudo-label generation speed, memory efficiency, and detection performance.

**Strengths:**

1. The solution proposed by the authors is well-described and well-supported by experimental results.
2. The paper is easy to follow.
3. The experiments are conducted four datasets of this field.
4. The paper is well-organized, with a clear flow from introduction to conclusion.

**Weaknesses:**

1. While the paper demonstrates the empirical effectiveness of PointOBB-v2, it could benefit from providing more theoretical justification or intuition behind the design choices, such as the label assignment strategy and the selection of hyperparameters.
2. The authors acknowledge that some hyperparameters, such as the radius in label assignment, are set based on the dataset. While this approach yields strong results on the tested datasets, it may require manual tuning when applied to new scenarios, potentially limiting the method's adaptability.
3.  Although the paper discusses the advantages of modular approaches' flexibility over end-to-end architectures, this method inherently constrains the dynamic updating of pseudo-labels during detector training—a crucial technique that has demonstrated significant efficacy in semi-supervised learning frameworks.

**Questions:**

1. Please explicitly present the objective function of CPM.
2.  The paper adopts an independent approach for pseudo-label generation and detector training, unlike the conventional semi-supervised teacher-student framework which enables dynamic pseudo-label updates. Please explain it.
3. While identical labels are assigned to positive samples of the same target. Why does the Class Probability Map (CPM) assign higher probabilities to regions surrounding the object centers and along their axes?
4. Please explain the term 'ignored' in Figure 2.
5. Please discuss the hyper parameters ( the radius and α ) in label assignment.

---

> ### Author Response · Authors · 2024-11-21
> **Feedback to Reviewer oGEA (1/2)**
>
> ## **Q1. The intuition behind the label assignment strategy.**
>
> **A1.** Our label assignment strategy is designed to ensure high certainty for both positive and negative samples. This means that samples classified as positive or negative have a high probability of being correctly labeled, while the "ignored" regions represent areas where positive and negative samples overlap. During training, the model tends to predict the entire object region up to its boundaries as the corresponding class in the CPM to minimize the loss. This allows the model to learn object contours instead of circular CPM patterns, despite differences between the training targets and predictions. The "ignored" label plays a crucial role in this, preventing the model from learning circular patterns.
> Regarding the hyperparameters in label assignment, we set the radius for positive samples as small as possible. For α\alphaα, we set it to 1, directly using the smallest radius for the circular assignment.
>
> ## **Q2. The advantages of modular approaches' flexibility.**
>
> **A2.** In the task of point-supervised oriented object detection, end-to-end methods like Point2RBox do not generate pseudo-labels directly; instead, they train the model to produce final outputs directly. In contrast, our method adopts a single-branch design that separates pseudo-label generation from detector training. While this approach does not support dynamic updating of pseudo-labels, it offers significant advantages in terms of flexibility. By decoupling pseudo-label generation, our method allows these labels to be reused across different detectors. This modularity enables seamless integration of newer or improved detectors without modifying the overall framework. On the other hand, end-to-end methods embed the detector within the model, making it difficult to replace or adapt, which greatly reduces their flexibility and limits experimental exploration.
>
> Additionally, we would like to clarify a potential misunderstanding regarding dynamic pseudo-label updates. Unlike semi-supervised object detection, weakly-supervised oriented object detection paradigms—such as ours—do not involve dynamic pseudo-label updates during training. In semi-supervised frameworks, dynamic pseudo-label updates are feasible because a small portion of the data includes accurate RBox annotations, allowing the teacher model to generate pseudo-labels with high accuracy from the outset. However, in point-supervised tasks, such as ours, only point annotations are available, and the lack of initial RBox annotations makes dynamic pseudo-label updates inapplicable. This distinction highlights the fundamental differences between semi-supervised and weakly-supervised frameworks.
>
> ## **Q3. Why higher probabilities surrounding object centers and along their axes.**
>
> **A3.** Thank you for pointing out this question. We explain this phenomenon in detail as follows:
>
> - **Surrounding object centers:** Positive samples are assigned within a small circle around the ground truth center, enabling the model to learn higher probabilities in these regions.
> - **Along their axes:** This characteristic often occurs with symmetric objects, such as sports courts, airplanes, and vehicles. In these cases, the axes exhibit less uncertainty when the model outputs high probabilities, which may be akin to symmetric learning. This suggests that the model is effectively capturing the inherent symmetry in object orientations.
>
> ## **Q4. What is the term 'Ignored' in Figure 2.**
>
> **A4.** "Ignored" indicates that these labels are excluded from training. This means that no matter what the CPM outputs at these points, they do not contribute to the loss calculation. This approach helps the model focus on regions with clear label assignments while avoiding ambiguous areas.

---

> ### Author Response · Authors · 2024-11-21
> **Feedback to Reviewer oGEA (2/2)**
>
> ## **Q5. The hyperparameters (radius and $\alpha$).**
>
> **A5.** We appreciate the concern about the sensitivity of thresholds. To address this, we conducted additional experiments:
>
> 1. Using the same thresholds as those applied to the DOTA dataset, we verified our method on the entirely different SKU110K-R retail dataset. Our method achieved a mAP$_{50}$ of 62.76, significantly outperforming Point2RBox's mAP$_{50}$ of 3.41. The poor performance of Point2RBox can be attributed to the large inter-class variability in SKU110K-R, which makes it challenging to generate suitable patterns for class representation. Additionally, PointOBB's SSC loss enforces inter-class consistency constraints, which inherently limits its applicability to single-class datasets like SKU110K-R. In contrast, our method remains effective in such scenarios, demonstrating its robustness and lower sensitivity to threshold parameters.
>
> 2. We conducted experiments on the DOTA dataset with varying parameters and observed minimal fluctuations in the final results. Additionally, the performance consistently exceeded that of PointOBB by a significant margin (about 8% gain on Oriented RCNN).
>   Therefore, we conclude that the introduced thresholds do not require precise tuning to achieve strong performance, indicating that the model is not highly sensitive to threshold settings.
>
> | Paramters Choice                             | mAP$_{50}$ (Oriented RCNN) |
> | -------------------------------------------- | -------------------------- |
> | $\alpha, b_1, b_2=1, 6, 4$ (result in paper) | 41.64                      |
> | $\alpha, b_1, b_2=0.67, 8, 4$                | 41.33                      |
> | $\alpha, b_1, b_2=1, 8, 4$                   | 40.68                      |
> | $\alpha, b_1, b_2=1.5, 8, 4$                 | 40.22                      |
> | $\alpha, b_1, b_2=1.5, 6, 4$                 | **41.95**                  |
> | $\alpha, b_1, b_2=0.67, 12, 6$               | 41.74                      |
>
> | Method      | DIOR-R | SKU110-R |
> | ----------- | ------ | -------- |
> | Point2RBox  | 27.34  | 3.41     |
> | PointOBB    | 38.08  | Not applicable in single class scenario       |
> | PointOBB-v2 | 39.62  | 62.76    |

---

> ### Author Response · Authors · 2024-11-25
>
> Dear Reviewers,
>
> May we kindly ask if our responses have addressed your concerns? We look forward to further discussions and feedback from you!
>
> Sincerely,
>
> Authors

---

> ### Comment · Area_Chair_nTg3 · 2024-11-25
> **review the rebuttal**
>
> Dear Reviewer oGEA,
>
> Could you kindly review the rebuttal thoroughly and let us know whether the authors have adequately addressed the issues raised or if you have any further questions.
>
> Best,
>
> AC of Submission1151

---

> ### Comment · Reviewer_oGEA · 2024-11-27
>
> Thanks to the authors for responding to most of my questions and the listed weakness. However, I notice that Q1 remains unanswered, and I've been waiting for a feedback on Q1.

---

> > ### Author Response · Authors · 2024-11-28
> >
> > Dear Reviewer:
> >
> > We sincerely apologize for not addressing Q1 in our initial rebuttal. Thank you for pointing this out, and we would like to provide a detailed response here.
> >
> > We first generate the training target for each point using our label assignment strategy. For positive and negative samples, we use focal loss to train the CPM. For ignored labels, we exclude them from the training process entirely—these points do not contribute to the loss function in any way.
> >
> > Thank you!
> >
> > Best regards,
> >
> > Authors

---

### Official Review · Reviewer_R8EQ · 2024-11-04

**Soundness:** 3
**Presentation:** 3
**Contribution:** 4
**Rating:** 8
**Confidence:** 5

**Summary:**

This paper presents PointOBB-v2, a novel approach for single point supervised oriented object detection, aimed at providing a simpler, faster, and stronger methodology compared to its predecessor, PointOBB. Specifically, PointOBB-v2 generates Class Probability Maps (CPM) using a positive and negative sample assignment strategy, which helps to capture the approximate contours of objects. To accurately estimate orientation and boundaries, Principal Component Analysis (PCA) is employed, providing a lightweight approach that avoids the need for prior deep network components or cumbersome teacher-student training frameworks. Experiments show that  PointOBB-v2 significantly improves pseudo-label generation efficiency, achieving a 15.58-fold increase in speed and reduced memory consumption while maintaining superior accuracy.

**Strengths:**

1. The primary contributions of PointOBB-v2 are its significant speed and efficiency improvements. The pseudo-label generation process is 15.58 times faster, and memory consumption is considerably reduced, requiring only around 8GB of GPU memory, compared to the previous version that faced significant out-of-memory issues. It also achieves higher accuracy on DOTA-v1.0/1.5/2.0 compared to PointOBB,  particularly in dense scenarios.
2. The experimental section of the paper demonstrates a comprehensive evaluation of PointOBB-v2, including comparisons across multiple versions of the DOTA dataset (v1.0, v1.5, and v2.0). The authors present quantitative results that highlight consistent improvements in accuracy and efficiency over the predecessor, PointOBB. The experimental setup effectively uses multiple detectors and datasets to validate the proposed method’s robustness and adaptability, particularly in dense object detection scenarios.
3. The manuscript is well-structured, with a logical flow that clearly explains the motivation, methodology, and outcomes. The authors effectively contextualize their work within the existing literature, highlighting the gap in current oriented object detection approaches and positioning PointOBB-v2 as an advancement in this domain.

**Weaknesses:**

1. The paper focuses primarily on aerial imagery datasets, and it is not entirely clear how well PointOBB-v2 would generalize to other domains such as autonomous vehicles or medical imaging. Although the aerial domain has its unique characteristics, a brief discussion or experiment on a different type of dataset could help demonstrate the method's broader applicability and flexibility.
2. There are still some grammatical issues in this paper that could be improved. In "Diverse from those approaches relying on one-shot samples," the phrase "Diverse from" is slightly awkward. Consider changing it to "Unlike approaches that rely on one-shot samples." Also, in "achieving significant improvements in the accuracy and generation speed of the pseudo-label, and improving memory efficiency," "improving" is repetitive here. Consider rewriting it as "achieving significant improvements in pseudo-label accuracy, generation speed, and memory efficiency."

**Questions:**

1. The paper mentions improved performance in high-density scenarios, but what specific attributes of PointOBB-v2 contribute most to these improvements?
2. Why was Principal Component Analysis (PCA) specifically chosen to estimate object orientation and boundary? Could other methods, such as clustering algorithms or deep learning-based orientation estimation, have been a better alternative? What are the trade-offs?

---

> ### Author Response · Authors · 2024-11-21
> **Feedback to Reviewer R8EQ**
>
> ## **Q1. The generalization of PointOBB-v2 to other domains.**
>
> **A1.** Thank you for pointing out the need to demonstrate broader applicability. As the following table shows, we conducted experiments in the retail scene using the SKU110-R dataset, where our method achieved an mAP$_{50}$ of 62.76. This result highlights the generalizability of PointOBB-v2 to domains beyond aerial imagery. In future work, we plan to further explore its application in other areas, such as autonomous vehicles and medical imaging.
>
> | Method      | DIOR-R | SKU110-R |
> | ----------- | ------ | -------- |
> | Point2RBox  | 27.34  | 3.41     |
> | PointOBB    | 38.08  | N/A in single class dataset |
> | PointOBB-v2 | 39.62  | 62.76    |
>
> ## **Q2. Grammatical issues and writing.**
>
> **A2.** We appreciate your constructive feedback on improving the writing quality. We will revise the paper to address the grammatical issues you highlighted. Specifically, we will rephrase "Diverse from those approaches relying on one-shot samples" to "Unlike approaches that rely on one-shot samples" and adjust "achieving significant improvements in the accuracy and generation speed of the pseudo-label, and improving memory efficiency" to "achieving significant improvements in pseudo-label accuracy, generation speed, and memory efficiency." Additionally, we will thoroughly review the manuscript to ensure grammatical consistency.
>
> ## **Q3. How model contribute to high-density scenarios.**
>
> **A3.** Our method excels in high-density object scenarios primarily due to two key components:
> - **Label assignment strategy (L259–266):** This ensures accurate and efficient sampling of positive and negative labels, even in dense object distributions.
> - **Vector constraint (L309–323):** This encourages precise orientation prediction while maintaining robustness in cluttered scenes.
>   These components collectively enhance the model's performance in high-density scenarios by addressing the unique challenges of such environments.
>
> ## **Q4. If any alternative methods for estimating orientation and boundary.**
>
> **A4.** Principal Component Analysis (PCA) was chosen for its simplicity, efficiency, and compatibility with weakly supervised learning. While other methods, such as the symmetric learning introduced in H2RBox-v2 [1], could potentially provide better orientation estimation, they may also introduce additional training complexity and time. Clustering algorithms might help in estimating boundaries, and we are considering exploring this direction in future work to further enhance the accuracy and flexibility of our method.
>
> [1] H2RBox-v2: Incorporating Symmetry for Boosting Horizontal Box Supervised Oriented Object Detection, NeurIPS, 2023

---

> > ### Comment · Reviewer_R8EQ · 2024-11-22
> > **Thank you**
> >
> > Thank you for the thorough response to my questions and listed weaknesses.

---

> > > ### Author Response · Authors · 2024-11-23
> > >
> > > Thank you for taking the time to review our responses. We greatly appreciate your constructive feedback and support.

---

### Official Review · Reviewer_qP7C · 2024-11-04

**Soundness:** 3
**Presentation:** 3
**Contribution:** 3
**Rating:** 8
**Confidence:** 5

**Summary:**

In this paper, PointOBB-v2 is proposed for point-supervised oriented object detection. The authors introduce the learning of Class Probability Map and use Principal Component Analysis to determine the size and rotation of objects. By employing these paradigms, the capability of the PointOBB-v2 is enhanced and significantly outperforms the PointOBB baselines.

**Strengths:**

1. The research topic of point-supervised oriented object detection is meaningful and challenging, and the paper is clear and well-organized.
2. As a new version of existing work (PointOBB), the comparison between them is essential. The experiments show the proposed method outperforms the PointOBB baselines in both training speed and accuracy by a considerable amount.

**Weaknesses:**

1. Since the authors use “the highest resolution feature map of the FPN”, it is not clear whether the resolution while learning the CPM affects the performance.
2. There are some minor issues: Please check Figure 2, Line 433, and Line 468. Some equations end with a period, while others end with a comma. Please ensure they are consistent.

**Questions:**

1. What does “the randomness introduced by sampling can cause variance in the result from a single pass” in Line 288 mean? Where does the randomness come from?
2. Table 2 shows that the improvement on the challenge datasets (DOTA-v1.5/v2.0) is more pronounced compared to DOTA-v1.0. What might account for this difference?

---

> ### Author Response · Authors · 2024-11-21
> **Feedback to Reviewer qP7C**
>
> ## **Q1. Whether the resolution while learning the CPM affects the performance.**
>
> **A1.** Thank you for raising this point. We use the highest resolution feature map of the FPN to ensure finer representation of orientation and boundary details. For example, a single pixel in the highest resolution CPM corresponds to $4 \times 4 = 16$ pixels in the original image. This level of detail is crucial because many object annotations in DOTA-v1.5 and DOTA-v2.0 are smaller than 10 pixels. Using the highest resolution feature map allows us to better capture these small objects, ensuring more precise orientation and boundary predictions.
>
> ## **Q2. Some typographical inconsistencies.**
>
> **A2.** We appreciate your attention to detail. We will address the typographical issues in Figure 2, Line 433, and Line 468, as well as ensure consistency in equation punctuation throughout the revised manuscript.
>
> ## **Q3. The "randomness" in Line 288.**
>
> **A3.** The "randomness" refers to the variance introduced during the PCA computation when sampling points based on their probabilities. Directly sampling points according to their pixel-level probabilities can result in variations across different passes, potentially disturbing the orientation estimation. To mitigate this, we compute a "weighted PCA," where probabilities are treated as weights rather than relying on stochastic sampling. This approach reduces variance and provides a more stable estimation of object orientation.
>
> ## **Q4. The more pronounced improvements on DOTA-v1.5/v2.0.**
>
> **A4.** The greater improvement on DOTA-v1.5 and DOTA-v2.0 can be attributed to the higher density of objects per image in these datasets. In such dense scenarios, methods like PointOBB face challenges due to the need to constrain the number of RoI proposals, leading to performance drops. Our approach, on the other hand, is robust to dense object distributions, allowing it to effectively handle the increased complexity of these datasets. This robustness underscores the strengths of our method in challenging scenarios.

---

> > ### Comment · Reviewer_qP7C · 2024-11-23
> >
> > The feedback has addressed my concerns. Due to the novel motivation, clear methodology, and comprehensive experimental analysis, I raised my rating. I suggest incorporating these modifications into the paper.

---

> > > ### Author Response · Authors · 2024-11-23
> > >
> > > Thank you for your thoughtful feedback and for raising your rating. We sincerely appreciate your recognition of our work's novelty, methodology, and experimental analysis. We will incorporate the suggested modifications into the revised paper to further improve its clarity and quality. Thank you again for your support!

---

### Official Review · Reviewer_GcxK · 2024-11-04

**Soundness:** 3
**Presentation:** 2
**Contribution:** 3
**Rating:** 6
**Confidence:** 4

**Summary:**

The paper presents PointOBB-v2, a faster and more accurate method for single-point supervised oriented object detection that improves upon PointOBB. By generating a Class Probability Map (CPM) with targeted sampling and applying Principal Component Analysis (PCA), PointOBB-v2 effectively estimates object orientation and boundaries without relying on pre-trained models. An added separation mechanism enhances performance in high-density scenarios. PointOBB-v2 achieves 15.58x faster training and significant accuracy gains on DOTA datasets, marking a new state-of-the-art in this domain.

**Strengths:**

## 1. Significant Contribution to Point-Supervised Oriented Object Detection

This paper presents a notable advancement in point-supervised oriented object detection, achieving improvements in both accuracy and training efficiency.

## 2. Comprehensive Experimental Analysis

The authors provide a thorough evaluation of the proposed method, including detailed ablation studies that effectively demonstrate its efficacy. The method is validated across three aerial image datasets, further underscoring its robustness and applicability.

**Weaknesses:**

## 1. Incomplete Explanation of the Class Probability Map (CPM)

- While the overall pipeline of PointOBB-v2 is generally understandable from L.198-211, the crucial workings of the CPM remain unclear.
- Specifically, key details about CPM training are omitted, including the training objective, activation function, and handling of ignored labels.
- The current structure of the Method section adds to the confusion. Section 3.1 describes CPM generation process, followed by Section 3.2, which outlines label assignment. (The first Section 3.1 assumes that we already have the completed CPM generation model, but the readers cannot understand how it can work.)
- Reordering these sections (such as by first describing the model training or merging both subsection) could enhance clarity.


## 2. Limited Justification for the Effectiveness of CPM

- The paper lacks convincing arguments for why the CPM functions effectively. For instance, the use of a fixed radius for positive and negative sampling lacks a clear rationale, particularly in how it helps define object contours.
- Sections 3.1 and 3.2 provide limited insight into how the model generates a CPM with distinct object regions.
- Figure 2, which illustrates the “Training Target” and the “Output CPM”, raises questions about how the model achieves such precise CPM output even when trained on such coarse training labels. Additional explanation on this discrepancy would be valuable.
- Further clarity on Figure 2 would be beneficial: is the red dot an output from the model or a manual illustration? Similarly, the meanings of colors in Figure 3 (blue, pink, and orange) should be clarified.

## 3. Generalization to Other Oriented Object Detection Scenarios
- While the method appears well-suited for densely packed object scenarios (e.g., aerial or retail images), it’s uncertain whether it generalizes to other tasks, such as text detection. Despite aiming to contribute to general object detection, the experiments are limited to remote sensing images, a subset of oriented object detection.


## Additional Comment

- The concept of CPM bears resemblance to the existing method TricubeNet [A], which, although fully supervised, also represents objects within a 2D map. A brief discussion of the similarities and distinctions between these approaches could enhance the paper’s context.

[A] TricubeNet: 2D Kernel-Based Object Representation for Weakly-Occluded Oriented Object Detection (WACV 2022)

**Questions:**

Please check the weaknesses.

## Justification

This paper presents notable advancements in point-supervised oriented object detection, but key limitations reduce its impact. The explanation of the CPM’s effectiveness is incomplete, particularly regarding how it generates clear object contours from coarse labels. While strong in experimental results, the paper would benefit from clearer methodology.

My initial rating is weak reject. I will finalize the rating after a discussion with the authors and other reviewers.

---

> ### Author Response · Authors · 2024-11-21
> **Feedback to Reviewer GcxK (1/2)**
>
> ## **Q1. Regarding the unclear explanation of CPM.**
>
> **A1.** We apologize for the lack of clarity in our original submission and provide a detailed explanation as follows:
>
>
> - **Training details, loss function of CPM** We appreciate your feedback and would like to clarify the training process. The **CPM training details** are introduced in L363–365: we train the model for 6 epochs using momentum SGD as the optimizer, with a weight decay of 1e-4. The initial learning rate is set to 0.005 and decays by a factor of 10 after the 4$^{th}$ epoch. The **activation function** used in the middle layers is LeakyReLU, while the last layer employs a sigmoid function. For each point, we first generate its target using a label assignment strategy. Subsequently, we train the CPM using **focal loss**.
> - **Ignored labels processing:** Ignored labels are excluded from the training process to ensure focus on well-defined samples.
> - **Structure order:** We apologize for the lack of clarity in the structure of the Method section. We will revise the paper to reorder the subsections and provide a smoother explanation. Specifically, we plan to first describe the model training process and then detail the label assignment strategy to improve the flow and understanding.
>
> ## **Q2. How CPM works.**
>
> **A2.** Thank you for your insightful feedback. We provide the following clarifications:
> - **Why the CPM learns object contours:** Our label assignment strategy does not rely solely on a fixed radius for sample assignment. While the positive samples are assigned using a fixed radius, the negative samples are determined using a variable radius strategy (L251–257). Specifically, for each GT object, we calculate the radius based on the distance to its nearest neighbor GT. This results in N radii, which are used to draw circles for negative sample assignment. The area outside these circles is designated as negative samples. Unlike fixed-length assignment, our method adapts to varying object densities, resulting in dynamic sample definitions. ***We also provide visualization results comparing our dynamic negative radius label assignment with the fixed negative radius approach. We found that when using a fixed radius, the CPM tends to learn object contours that are more circular, with the CPM primarily concentrated around the object's center. In contrast, our method produces clearer object boundaries. Please see the Appendix in our revised paper for detailed visualizations.***
>
>   The design of the positive and negative sample definitions ensures high certainty in their classification. Ignored regions, representing areas where positive and negative samples overlap, allow the model to focus on clear distinctions between object and non-object regions. This inherently guides the model to learn object contours, as expanding predictions to the true boundary minimizes the loss. The ignored labels play a critical role in preventing the model from learning circular CPM patterns, thereby enabling contour learning.
> - **Training target vs. output CPM:** The differences between the training target and the output CPM have been addressed above, highlighting how the model effectively learns object contours despite coarse supervision.
> - **Clarification of Figures 2 and 3:** We apologize for the lack of clarity in the figures. In Figure 2, the red point represents the ground truth point annotation. In Figure 3, blue, pink, and orange regions denote the highest CPM values for the classes harbor, ship, and large-vehicle, respectively.
>
> ## **Q3. Regarding generalization to other scenarios.**
>
> **A3.** Indeed, our method performs exceptionally well in dense object scenarios. To further demonstrate this, we conducted additional experiments on the SKU110K-R dataset, achieving a remarkable mAP$_{50}$ of 62.76, which is significantly higher than Point2RBox's 3.41. The poor performance of Point2RBox can be attributed to the large inter-class variability in SKU110K-R, making it difficult to generate suitable patterns. Additionally, PointOBB's SSC loss enforces inter-class consistency constraints, which inherently limits its applicability to single-class datasets like SKU110K-R. The experimental results are provided in the table below.
>
> However, we must also acknowledge that our method performs less effectively on text detection tasks. From our experimental analysis, we observed that the spacing between characters and gaps within individual characters make CPM sample assignment less effective. Even when employing the smallest radius, positive samples inevitably include background regions, reducing detection accuracy. We will include this finding in the limitations section of the paper. Thank you for your professional feedback and constructive suggestions.
>
> | Method      | DIOR-R | SKU110-R |
> | ----------- | ------ | -------- |
> | Point2RBox  | 27.34  | 3.41     |
> | PointOBB    | 38.08  | N/A in single class dataset |
> | PointOBB-v2 | 39.62  | 62.76    |

---

> ### Author Response · Authors · 2024-11-21
> **Feedback to Reviewer GcxK (2/2)**
>
> ## **Q4. Comparison to TricubeNet.**
>
> **A4.** Thank you for pointing out the resemblance to TricubeNet. We will add a discussion in Section 3.1 of the revised paper to highlight the differences between our CPM and TricubeNet. Specifically:
>
> - TricubeNet uses a symmetric 2D kernel representation, which might offer advantages for precise angle prediction.
> - Our CPM, while not symmetric, employs a weakly supervised approach, enabling it to capture approximate object contours with minimal annotation effort.

---

> ### Author Response · Authors · 2024-11-25
>
> Dear Reviewers,
>
> May we kindly ask if our responses have addressed your concerns? We look forward to further discussions and feedback from you!
>
> Sincerely,
>
> Authors

---

> ### Comment · Area_Chair_nTg3 · 2024-11-25
> **review the rebuttal**
>
> Dear Reviewer GcxK,
>
> Could you kindly review the rebuttal thoroughly and let us know whether the authors have adequately addressed the issues raised or if you have any further questions.
>
> Best,
>
> AC of Submission1151

---

> > ### Comment · Reviewer_GcxK · 2024-11-26
> >
> > I appreciate the authors for addressing all my concerns.
> > No other questions.
> > I will finalize the rating after considering all the reviews and comments from other reviewers.
> >
> > Thanks.

---

> > > ### Author Response · Authors · 2024-11-26
> > >
> > > Thank you for your thoughtful feedback and for taking the time to review our responses. We appreciate your consideration and are happy to address any further questions if needed.

---

> > > ### Author Response · Authors · 2024-12-01
> > >
> > > Dear Reviewer,
> > >
> > > Thank you once again for your valuable feedback and for your willingness to reassess your rating. Most reviewers have already provided their feedback and indicated that we have addressed their concerns. As the discussion deadline is approaching, we would like to kindly send a reminder about the reassessment and inquire if there is anything further we can clarify or address.
> > >
> > > We deeply appreciate your time and effort in evaluating our work, and we are truly grateful for your constructive feedback.
> > >
> > > Thank you once again for your support and consideration.
> > >
> > > Best regards,
> > >
> > > Authors

---

> > > > ### Comment · Reviewer_GcxK · 2024-12-02
> > > >
> > > > After checking all the reviews and rebuttals, I raised the rating.
> > > > I strongly recommend that the authors improve the final version based on the reviews.

---

> > > > > ### Author Response · Authors · 2024-12-02
> > > > >
> > > > > Dear Reviewer,
> > > > >
> > > > > Thank you very much for your thoughtful reconsideration and for raising the rating. We deeply appreciate your valuable feedback and will certainly take all the reviews into account as we improve the final version of the paper.
> > > > >
> > > > > Thank you again for your support!
> > > > >
> > > > > Best regards,
> > > > >
> > > > > Authors

---

### Official Review · Reviewer_ND69 · 2024-11-04

**Soundness:** 4
**Presentation:** 4
**Contribution:** 3
**Rating:** 8
**Confidence:** 4

**Summary:**

In this paper, the authors propose a single point supervised oriented object detection method called PointOBB-v2, which can generate pseudo rotated boxes from points without relying on any other prior. Compared with PointOBB, PointOBB-v2 introduces a novel pipeline to improve the accuracy and generation speed of the pseudo-label. Extensive experiments on the DOTA-v1.0/v1.5/v2.0 demonstrate the effectiveness of the proposed method.

**Strengths:**

1. The paper is innovative, well-organized, and well-written.
2. The proposed method is an improvement of the structure of PointOBB.
3. The authors conduct comprehensive experiments to demonstrate the effectiveness of the proposed method.

**Weaknesses:**

1. In related work, it is better to provide a summary of the methods described in section 2.1.
2. The authors conduct experiments on DOTA datasets. How about the effectiveness of the proposed method on other datasets with different object categories and data distributions, such as DIOR?

**Questions:**

See the Weaknesses section.

---

> ### Author Response · Authors · 2024-11-21
> **Feedback to Reviewer ND69**
>
> ## **Q1. In related work, it is better to provide a summary of the methods described in section 2.1.**
>
> **A1.** Thank you for your feedback on improving the clarity of Section 2.1. In response, we have revised the related work section to provide a concise summary of existing methods.
>
> **Two-stage methods** such as Oriented R-CNN, RoI Transformer, and ReDet have introduced mechanisms to handle scale, aspect ratio variations, and rotation explicitly. These methods emphasize alignment features or convolutional adaptations to improve detection accuracy in dense and cluttered environments.
>
> **Single-stage methods** like R$^3$Det, S$^2$ANet, and DAL focus on improving efficiency without sacrificing precision. Strategies include novel loss functions such as GWD, KLD, and KFIoU, as well as methods for boundary-free angle regression and Gaussian-based loss designs to address discontinuity issues.
>
> In addition to these, RepPoint-based methods predict a set of sample points to describe object extents, providing an alternative to traditional box-based approaches. Methods such as ARC and SASM modify convolutional layers to better handle rotation, demonstrating the evolving strategies for efficient and robust detection.
>
> ## **Q2. How about the effectiveness of the proposed method on other datasets with different object categories and data distributions, such as DIOR?**
>
> **A2.** To further validate the generalizability of our method, we conducted experiments on DIOR-R and SKU110-R. The results, presented in following table, demonstrate that our method consistently improves performance on datasets with diverse object categories and distributions. This indicates the broader applicability and robustness of our approach beyond the DOTA dataset. For more analysis, please refer to **Reviewer TgZt's Q2/A2**
>
> | Method      | DIOR-R | SKU110-R |
> | ----------- | ------ | -------- |
> | Point2RBox  | 27.34  | 3.41     |
> | PointOBB    | 38.08  | N/A in single class dataset |
> | PointOBB-v2 | 39.62  | 62.76    |

---

> > ### Comment · Reviewer_ND69 · 2024-11-25
> >
> > My concerns are addressed now. I will keep the original rating.

---

> > > ### Author Response · Authors · 2024-11-25
> > >
> > > Thank you for your feedback and for taking the time to review our responses. We truly appreciate your thoughtful review and support of our work!

---

### Official Review · Reviewer_TgZt · 2024-11-04

**Soundness:** 3
**Presentation:** 2
**Contribution:** 3
**Rating:** 6
**Confidence:** 3

**Summary:**

This paper improves PointOBB, an existing weakly supervised oriented object detection method. Oriented object detection is important for analyzing aerial and satellite imagery. The cost of annotation, especially when you want to draw oriented boxes, is high. This paper (and, previously, PointOBB) proposes to annotate an object with just a single point, which is an easier annotation method. The proposed method first generates a class probability map, which creates positive and negative regions. Then, a PCA based analysis is done to estimate principal axes of objects to generate rotated boxes as pseudo annotation. Authors report improved performance over PointBB on the DOTA datasets.

**Strengths:**

Performance improvement over the baseline is impressive. Also, the proposed method is 15x faster to train.

**Weaknesses:**

The proposed method (i) is a simple increment to PointOBB and (ii) involves many thresholds, indicating that it might not be an easily generalizable one:

(i) The method boils down to class probability map (CPM) and PCA.  CPM is a simple positive/negative segmentation.

(ii) The method has at least 6 thresholds: radius b1 (L244), radius b2 (L262), multiplier to radius (L254), and three others at lines 209, 307 and 320. This suggests that the method is fragile and not easily generalizable to new datasets. I believe it is the authors' responsibility to show that their method is generalizable. Yet, results are reported only on the DOTA dataset.


The main claim of the paper is that they can do oriented object detection with weak supervision (single point per object). This is good but the performance (around 45)  is drastically behind the fully supervised methods (https://paperswithcode.com/sota/oriented-object-detection-on-dota-1-0), which is around high 70s. So, I think, it  is reasonable to expect a discussion on full annotation cost vs performance degradation. However, none is provided.

**Questions:**

- What are the annotation costs for full annotation and single point? How do they compare? Is low performance worth the reduced annotation  cost?
- How does PointOBBv2 perform on DIOR-R and other oriented object detection datasets?


## Post-rebuttal edit:

The authors provided satisfactory answers to my questions above. They promise to include these answers in the final version of the paper.

---

> ### Author Response · Authors · 2024-11-21
> **Feedback to Reviewer TgZt (1/2)**
>
> ## **Q1. The proposed method (i) is a simple increment to PointOBB.**
>
> **A1.** We apologize if the paper has caused any misunderstanding. Here, we would like to provide further clarification. PointOBB is based on multiple instance learning (MIL) and symmetric learning, while PointOBB-v2 relies on class probability maps (CPM) and principal component analysis (PCA). Therefore, PointOBB and PointOBB-v2 follow entirely different technical paradigms. Specifically，unlike PointOBB, which uses a computationally expensive dual-branch structure and enforces consistency across various image transformations, our method utilizes a streamlined single-branch design. By leveraging a carefully crafted positive-negative sample assignment strategy, we train the CPM and subsequently use weighted PCA to infer the orientation. This innovative approach is both simple and effective, leading to improved detection accuracy while significantly reducing time and memory consumption.
>
> Moreover, some reviewers have acknowledged our novelty and highlighted that our work introduces an innovative pipeline compared to PointOBB. For example, reviewer ND69 mentions that "PointOBB-v2 introduces a novel pipeline," reviewer R8EQ states that "This paper presents PointOBB-v2, a novel approach for single point supervised oriented object detection, ..., providing a lightweight approach that avoids the need for prior deep network components or cumbersome teacher-student training frameworks," and reviewer oGEA highlights that "the authors introduce PointOBB-v2, which presents a streamlined pipeline that eliminates the traditional teacher-student architecture."
>
> **Thus, we believe this work represents a meaningful contribution, rather than a marginal improvement.**
>
> ## **Q2. (ii) involves many thresholds, indicating that it might not be an easily generalizable one.**
>
> **A2.** We appreciate the concern about the sensitivity of thresholds. To address this, we conducted additional experiments:
>
> 1. Using the same thresholds as those applied to the DOTA dataset, we verified our method on the entirely different SKU110K-R retail dataset. Our method achieved a mAP$_{50}$ of 62.76, significantly outperforming Point2RBox (mAP$_{50}$ of  3.41). The poor performance of Point2RBox can be attributed to the large inter-class variability in SKU110K-R, which makes it challenging to generate suitable patterns for class representation. Additionally, PointOBB's SSC loss enforces inter-class consistency constraints, which inherently limits its applicability to single-class datasets like SKU110K-R. In contrast, our method remains effective in such scenarios, demonstrating its robustness and lower sensitivity to threshold parameters.
>
> | Method      | DIOR-R | SKU110-R |
> | ----------- | ------ | -------- |
> | Point2RBox  | 27.34  | 3.41     |
> | PointOBB    | 38.08  | N/A in single class dataset       |
> | PointOBB-v2 | 39.62  | 62.76    |
>
> 2. We conducted experiments on the DOTA dataset, focusing on parameters that could potentially impact the results ($\alpha$, b1, b2), and observed minimal fluctuations in the final results. Additionally, the performance consistently exceeded that of PointOBB (33.31%) by a significant margin (minimum 7% gain on Oriented RCNN).
>   Therefore, we conclude that the introduced thresholds do not require precise tuning to achieve strong performance, indicating that the model is not highly sensitive to threshold settings.
>
> | Paramters Choice                             | mAP$_{50}$ (Oriented RCNN) |
> | -------------------------------------------- | -------------------------- |
> | $\alpha, b_1, b_2=1, 6, 4$ (result in paper) | 41.64                      |
> | $\alpha, b_1, b_2=0.67, 8, 4$                | 41.33                      |
> | $\alpha, b_1, b_2=1, 8, 4$                   | 40.68                      |
> | $\alpha, b_1, b_2=1.5, 8, 4$                 | 40.22                      |
> | $\alpha, b_1, b_2=1.5, 6, 4$                 | **41.95**                  |
> | $\alpha, b_1, b_2=0.67, 12, 6$               | 41.74                      |

---

> ### Author Response · Authors · 2024-11-21
> **Feedback to Reviewer TgZt (2/2)**
>
> ### **Q3. The paper shows weakly supervised detection but lags in performance (45 vs 70+) without cost-performance discussion.**
>
> **A3.** Thank you for raising this important question. We agree that it is necessary to discuss the trade-offs between full annotation cost and performance degradation, and we provide the following explanations:
>
> 1. In the field of object detection, the cost of point annotations is about 36.5% lower than HBB and 104.8% lower than OBB, and the efficiency of labeling point is significantly higher than both HBB and
> OBB, refer to [google annotation platform pricing](https://cloud.google.com/ai-platform/data-labeling/pricing). This cost analysis has also been adopted and validated in prior works, such as H2RBox，PointOBB and Point2RBox, further demonstrating its reliability and relevance.
>
> 2. Although the performance of point supervision still lags behind that of full supervision, our method takes a significant step forward compared to existing approaches such as PointOBB, achieving improvements of 10.90/25.15/21.19 mAP$_{50}$ on the DOTA-v1.0/1.5/2.0 datasets. We believe that future research can further narrow the gap, and this work provides a strong example to build upon.
>
> ### **Q4. Regarding the suitability of the work for ICLR (representation learning).**
>
> **A4.** We believe that our submitted work aligns with the scope of ICLR for the following reasons:
>
> 1. ICLR specifies the following primary areas for submission: "applications to computer vision, audio, language, and other modalities" and "unsupervised, self-supervised, semi-supervised, and supervised representation learning." We consider our work to be consistent with both these areas.
>
> 2. Weakly supervised oriented object detection algorithm H2RBox [1] and fully supervised oriented object detection algorithm KFIoU [2] were both accepted at ICLR 2023, demonstrating the relevance of oriented object detection to ICLR.
>
> 3. This year's ICLR submissions also include a significant number of papers related to oriented and general object detection.
>
> [1] H2RBox: Horizontal Box Annotation is All You Need for Oriented Object Detection, ICLR, 2023
>
> [2] The KFIoU Loss for Rotated Object Detection, ICLR, 2023
>
>
> ### **Q5. How does PointOBBv2 perform on DIOR-R and other oriented object detection datasets?**
>
> **A5.** To further validate the generalizability of our method, we conducted experiments on DIOR-R and SKU110-R. The results, presented in following table, demonstrate that our method consistently improves performance on datasets with diverse object categories and distributions. This indicates the broader applicability and robustness of our approach beyond the DOTA dataset.
>
> | Method      | DIOR-R | SKU110-R |
> | ----------- | ------ | -------- |
> | Point2RBox  | 27.34  | 3.41     |
> | PointOBB    | 38.08  | N/A in single class dataset |
> | PointOBB-v2 | 39.62  | 62.76    |

---

> ### Author Response · Authors · 2024-11-25
>
> Dear Reviewers,
>
> May we kindly ask if our responses have addressed your concerns? We look forward to further discussions and feedback from you!
>
> Sincerely,
>
> Authors

---

> ### Comment · Area_Chair_nTg3 · 2024-11-25
> **review the rebuttal**
>
> Dear Reviewer TgZt,
>
> Could you kindly review the rebuttal thoroughly and let us know whether the authors have adequately addressed the issues raised or if you have any further questions.
>
> Best,
>
> AC of Submission1151

---

> ### Comment · Reviewer_TgZt · 2024-11-25
>
> Dear authors,
>
> Thank you for addressing all my questions and concerns. I will re-assess. I do not have any further questions.

---

> > ### Author Response · Authors · 2024-11-25
> >
> > Thank you for taking the time to review our responses. We greatly appreciate your constructive feedback and support. If you have any additional questions or thoughts, we would be happy to address them.

---

> > ### Author Response · Authors · 2024-12-01
> >
> > Dear Reviewer,
> >
> > Thank you once again for your valuable feedback and for your willingness to reassess your rating. Most reviewers have already provided their feedback and indicated that we have addressed their concerns. As the discussion deadline is approaching, we would like to kindly send a reminder about the reassessment and inquire if there is anything further we can clarify or address.
> >
> > We deeply appreciate your time and effort in evaluating our work, and we are truly grateful for your constructive feedback.
> >
> > Thank you once again for your support and consideration.
> >
> > Best regards,
> >
> > Authors

---

> > > ### Comment · Reviewer_TgZt · 2024-12-01
> > >
> > > Dear authors,
> > >
> > > Are you planning to add (i) the results on other datasets and (ii) discussion on annotation cost vs performance gap to full-supervision accuracy to the paper? I think your answers to these questions here are satisfactory but the paper in its current form does not include these important points, thereby providing an incomplete picture to the reader.
> > >
> > > Best,

---

> > > > ### Author Response · Authors · 2024-12-01
> > > >
> > > > Dear Reviewer,
> > > >
> > > > Thank you very much for your insightful feedback and for raising these important points. We sincerely appreciate your careful consideration of our work!
> > > >
> > > > We would like to assure you that the results on additional datasets and the discussion on annotation cost vs. performance gap will certainly be included in the final version of the paper. As the current PDF submission has already been finalized and cannot be modified, we will ensure these aspects are fully addressed in the camera-ready version if our paper is accepted.
> > > >
> > > > Thank you once again for your valuable suggestions and we greatly appreciate your support!
> > > >
> > > > Best regards,
> > > >
> > > > Authors

---

### Author Response · Authors · 2024-11-21
**To all reviewers**

**We sincerely thank all the reviewers for providing us with an abundance of valuable feedback, which is critical to improving our paper. We are pleased to learn that the novelty of our work has been acknowledged (as noted by reviewers oGEA, ND69, and R8EQ), the impressive improvements over previous state-of-the-art methods have been recognized (acknowledged by all reviewers), the completeness of our experiments has been affirmed (highlighted by reviewers ND69, GcxK, R8EQ, and oGEA), and the clarity of our writing has been appreciated (recognized by Reviewers oGEA, R8EQ, qP7C, and ND69). We will make every effort to address each reviewer’s questions thoroughly in our rebuttal. Thank you again for your support of our work, and we deeply appreciate the effort each reviewer has dedicated in reviewing our paper.**

---

### Meta-Review · Area_Chair_nTg3 · 2024-12-17

**Metareview:**

(a) This paper introduces PointOBB-v2, a single-point supervised oriented object detection method that generates pseudo rotated boxes directly from point annotations without additional priors. It presents a notable advancement in point-supervised oriented object detection, achieving improvements in both accuracy and training efficiency.

(b) The strengths of the paper are 1) well-organized, well-written writings; 2) significant speed and efficiency improvements (e.g., 15x faster generation speed and reduced memory, only 8GB); 3) sufficient comparisons and experiments which provides a thorough comparison with the original PointOBB.

(c) The first version of this paper involves extensive threshold tuning, lacks experiments on other datasets and discussions of related work, and contains some typos and issues with label assignments. These weaknesses are effectively addressed during the rebuttal.

(d) The most important reasons to accept this paper are that the authors push the boundary of the topic of ``point-supervised oriented object detection'', achieving significant speed and efficiency improvements, making the technique more practical.

**Additional Comments On Reviewer Discussion:**

(a) Reviewer TgZt highlights concerns about threshold sensitivity and the lack of discussion on the trade-off between full annotation cost and performance degradation. The authors address these issues by conducting additional experiments, showing that threshold selection is not sensitive. They further demonstrate that point annotations reduce costs by approximately 36.5% compared to HBB and 104.8% compared to OBB, arguing that this work establishes a strong baseline for future efforts to bridge the gap between full and point annotations.

(b) Reviewer ND69 suggests summarizing methods in Section 2.1 for clarity. They also recommend testing the proposed method on datasets with varied categories and distributions, like DIOR, to evaluate its effectiveness further. The authors successfully address the issues.

(c) Reviewer GcxK raises concerns about the incomplete explanation and limited justification of the Class Probability Map (CPM), as well as its generalization to other oriented object detection scenarios. The authors adequately address these issues, leading the reviewer to raise their score following the rebuttal period.

(d) Reviewer qP7C questions whether using the highest resolution feature map of the FPN impacts CPM performance. They also note minor issues with Figure 2, Line 433, Line 468, and inconsistent punctuation in equations, suggesting corrections for consistency. The authors successfully address the issues.

(e) Reviewer R8EQ notes that the paper focuses on aerial imagery and suggests testing PointOBB-v2 on other domains to demonstrate broader applicability. They also point out grammatical issues, recommending clearer phrasing and avoiding repetition for improved readability. The authors successfully address the issues.

(f) Reviewer oGEA finds the paper shows strong empirical results but lacks theoretical justification for key design choices like label assignment and hyperparameters. Some hyperparameters are dataset-specific, which may require manual tuning for new scenarios, limiting adaptability. Additionally, the method's modular design restricts dynamic pseudo-label updates, a key technique in semi-supervised learning. The initial rebuttal addresses most concerns, but Q1 remains unanswered. After further clarification from the authors, the reviewer decides to maintain their score.

---

### Decision · Program_Chairs · 2025-01-22

Accept (Poster)